# Interlingual Interactions Elicit Performance Mismatches Not "Compromise" Categories in Early Bilinguals: Evidence from Meta-Analysis and Coronal Stops

**Joseph V. Casillas** 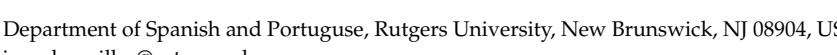

Department of Spanish and Portuguse, Rutgers University, New Brunswick, NJ 08904, USA;
joseph.casillas@rutgers.edu

**Abstract:** Previous studies attest that some early bilinguals produce the sounds of their languages in a manner that is characterized as "compromise" with regard to monolingual speakers. The present study uses meta-analytic techniques and coronal stop data from early bilinguals in order to assess this claim. The goal was to evaluate the cumulative evidence for "compromise" voice-onset time (VOT) in the speech of early bilinguals by providing a comprehensive assessment of the literature and presenting an acoustic analysis of coronal stops from early Spanish–English bilinguals. The studies were coded for linguistic and methodological features, as well as effect sizes, and then analyzed using a cross-classified Bayesian meta-analysis. The pooled effect for "compromise" VOT was negligible ($\beta = -0.13$). The acoustic analysis of the coronal stop data showed that the early Spanish–English bilinguals often produced Spanish and English targets with mismatched features from their other language. These performance mismatches presumably occurred as a result of interlingual interactions elicited by the experimental task. Taken together, the results suggest that early bilinguals do not have "compromise" VOT, though their speech involves dynamic phonetic interactions that can surface as performance mismatches during speech production.

**Keywords:** compromise VOT; voice timing; bilingualism; performance mismatches; dynamic phonetic interactions

## 1. Introduction

Though early bilinguals with ample experience in their first (L1) and second (L2) languages are believed to show "monolingual-like" L2 speech production (Rao and Ronquest 2015), research on bilingual language modes (Grosjean 2001) has shown that cross-linguistic interactions are strengthened in bilingual contexts (Olson 2013; Simonet 2014) and can lead to production/perception that differs from that of monolinguals. A crucial question revolves around how a bilingual speaker mitigates producing/perceiving acoustically similar segments in their languages in unilingual and bilingual settings, i.e., whether the two systems are kept separate (see Magloire and Green 1999), or whether there is a compromise in the acoustic characteristics of the sounds of the two languages (see Caramazza et al. 1973). A body of research dating back to the 1970s shows that some early bilinguals produce the sounds of their languages in a manner that has been characterized as "compromise", "intermediate" or "merged" with regard to monolingual distributions (e.g., Flege and Hillenbrand 1984; Flege 1991; Flege and Eefting 1987b; Sundara et al. 2006, among many others). The present study takes a systematic look at the cumulative evidence for "compromise" speech production in early simultaneous and sequential bilinguals. This study begins by describing the nature of "compromise" categories and reviews the literature supporting their existence. Next, this study considers how "compromise" categories may arise under current models of bilingual phonology, and then describes some methodological concerns. Afterwards, this study employs meta-analytic techniques to assess the extant literature,

and concludes by presenting an alternative account of "compromise" categories using production data from coronal stops.

## 1.1. Background and Motivation

A recurring finding in the bilingual speech literature is that even early bilinguals, both simultaneous and sequential, can display production, perception, and lexical processing that differs from monolinguals.[1] To be more specific, this finding is often framed in terms of "compromise" or "intermediate" phonetic categories in the bilingual production of stop contrasts, mainly because the acoustic properties of the segments are proposed to lie somewhere between those of the two languages. In the present work, a "compromise" category is defined as one that is not target-like with regard to some acoustic property. This particular line of research has focused on voice timing. Voice timing in stops can be determined by a number of parameters, the most common of which is voice-onset time (VOT, Lisker and Abramson 1964). VOT refers to the duration of the interval between the release of the stop and the onset of voicing of the following segment. VOT realizations include phonetically voiced stops, in which voicing begins before the release (i.e., lead VOT), as well as phonetically voiceless, lag stops, in which voicing begins after the release (i.e., short- and long-lag VOT, approx. 0–30 ms and 30+ ms, respectively). Many languages have two-way or three-way oppositions in which lead VOT, short-lag VOT, and long-lag VOT are mapped to phonologically voiced and voiceless categories. French, for example, contrasts voiced and voiceless stops with lead VOT and short-lag VOT, respectively. English, on the other hand, contrasts voiced and voiceless stops with short- and long-lag VOT, respectively.

The notion of "compromise" categories appears to have been coined by Williams (1980) in reference to Williams (1977), a study on Spanish–English bilinguals' production of bilabial stops. Williams (1980) stated, "The apparent development of compromise VOT targets in both perception and production for bilinguals and second-language learners may reflect a true convergence over time of the acoustic phonetic features of the two languages instead of the development of two separate phonetic systems" (p. 213). Subsequently, the term became commonplace in the bilingual speech literature (e.g., Flege and Hillenbrand 1984; Antoniou et al. 2010, 2011; Bullock et al. 2006; Chang et al. 2011; Flege 1991; Flege and Eefting 1987b, 1988; Gabriel et al. 2016; Jones 2020; Kehoe et al. 2004; Kiliç 2018; Kilpatrick 2004; Lein et al. 2016; Llama and López-Morelos 2016; López 2012; Morgan 2011; Sundara et al. 2006).

One explanation for the notion that bilinguals display "compromise" categories is offered by the Speech Learning Model (SLM, Flege 1995). The SLM posits that the ability to learn speech sounds is maintained throughout life. Novel phones are stored in long-term memory as phonetic categories. Importantly, L1 and L2 phonetic categories interact because they are assumed to share the same phonological space. L2 sounds are unconsciously and automatically linked with neighboring L1 phonetic categories via a mechanism Flege (1995) refers to as the "equivalence classification". According to the equivalence classification, linked phones that are perceived as being phonetically similar may result in a single, merged category used for both languages. If this is the case, the model posits that the L1 sound may assimilate to the L2 sound, resulting in both L1 and L2 productions that are intermediate in the phonetic space with regard to monolingual categories. If, on the other hand, the linked phones are perceived as being phonetically dissimilar, it is more likely that a new phonetic category will be formed. In this situation, the model predicts that the L2 category can dissimilate from nearby sounds in order to maintain phonetic contrast. Thus both scenarios can lead to "compromise" categories in bilingual speech.

Though they did not use the term "compromise" VOT, evidence for the phenomenon in early bilinguals dates back to a landmark study by Caramazza et al. (1973). This study

---

1. The present work takes a broad view, similar to that of the studies referenced herein, on what constitutes early bilingualism. Specifically, an early bilingual is operationalized as one who was exposed to an additional language before the age of 12. This study also distinguishes between simultaneous and sequential bilingualism, where the former refers to an individual that acquires their languages at the same time and the latter refers to an individual that acquires an L2 after the L1.

examined voiceless stop production/perception in early French–English bilinguals that had acquired English before the age of seven. Caramazza et al. (1973) compared the bilinguals' performance on a perceptual identification task and a reading task with monolingual English and French speakers and found that, when the bilinguals were in French mode, they produced French /ptk/ no differently than the French monolingual control group. However, when they were tested in English mode, they produced English /ptk/ with less aspiration than the monolinguals. Caramazza et al. (1973) concluded that French–English bilinguals were "more closely aligned" with the French monolinguals, presumably because they had learned English sequentially.

The results from Caramazza et al. (1973) suggest that sequential learners that consistently use both languages over a long period of time may still produce stops in a way that differs from monolingual speech. An SLM account for these data might propose that category formation was blocked because of the equivalence classification. This would imply that the phones were still linked, though the bilinguals' L1 categories did not differ from those of monolingual controls, and, therefore, did not assimilate or merge.

In another seminal study in this literature, Flege and Eefting (1987b), examined the production of Spanish stops in numerous groups of Puerto Rican Spanish–English sequential bilingual children and adults with different linguistic backgrounds. Spanish stops contrast lead VOT with short-lag VOT similar to French. The bilinguals were compared with age-matched monolingual controls for both languages. Of particular relevance to the present work are two early bilingual groups that Flege and Eefting (1987b) referred to as earlier childhood bilinguals (ECB) and later childhood bilinguals (LCB). Both the ECB and LCB groups comprised adults who had learned English before the age of seven, but the ECB group was born in the U.S. mainland, or moved there shortly after birth, whereas the LCG group still lived in Puerto Rico. Both groups produced English /ptk/ with less aspiration than monolingual English speakers. Flege and Eefting (1987b) concluded that the early bilinguals—even those living in the U.S. since early childhood—were not able to produce English /ptk/ "authentically", suggesting that their "intermediate" productions lent support to the equivalence classification hypothesis. In other words, they continued to associate the Spanish and English stops as realizations of the same phonetic category.

The results from Caramazza et al. (1973) and Flege and Eefting (1987b) suggest that sequential learners will differ from monolinguals because of the equivalence classification, but "compromise" VOT has also been documented in simultaneous bilinguals (see Sundara et al. 2006; Fowler et al. 2008; Kupisch and Lleó 2017; Lein et al. 2016, among others). For instance, in a more recent study, Fowler et al. (2008) analyzed the voiceless stops of English–French simultaneous and early, sequential bilinguals by comparing their production with monolingual controls. The simultaneous bilinguals produced French /ptk/ with longer VOT than monolingual French speakers, and English /ptk/ with shorter VOT than monolingual English speakers. The effects were greater in the early, sequential bilinguals who produced stops that were characterized as even more intermediate in both languages. Though the SLM was designed with sequential learners in mind, Fowler et al. (2008) proposed that, for the simultaneous bilinguals, the different phones were not merged but still "cognitively identified with one another" because of the equivalence classification (p. 650).

The "compromise" VOT literature also shows significant between-study variability, as there are also studies that do not find "compromise" categories in bilingual production. For instance, Flege (1991) analyzed early and late Spanish–English bilinguals' production of Spanish and English /t/ in utterance initial and utterance medial position. Importantly, the voiceless coronal is produced with short-lag VOT in Spanish and long-lag VOT in English. Flege (1991) found that, in both positions, the late bilinguals produced English /t/ with "compromise" VOT, but the early bilinguals' production was no different from the monolingual speakers. Thus, it seems that some early bilinguals are able to establish se-parate phonetic categories and maintain separation between their languages, even when the segments are acoustically similar. Nevertheless, bilinguals do indeed seem to display

more variability in their production of stops. For example, a recurring finding is that bilinguals whose L1 is a 'true voicing' language, that is, one that contrasts pre-voiced stops with short-lag stops (i.e., Spanish, French), tend to produce English voiced stops with pre-voicing at higher rates than monolingual English speakers (e.g., MacLeod and Stoel-Gammon 2005; Flege and Eefting 1987b; Hazan and Boulakia 1993). Much like in monolingual stop production (Chodroff and Wilson 2017), bilingual stop production also shows structured co-variation between stop categories (Chodroff and Baese-Berk 2019). This raises the possibility that variable voiced stop production might also be structured in other ways, on dimensions other than VOT.

### 1.2. Motivating Meta-Analysis

Given the discrepancies in the literature regarding the reliability of "compromise" categories, it is imperative that one consider all possible alternative explanations. One notable issue regarding voice timing that bares on this line of research is related to the mea-sure of VOT itself. Manifold studies show that VOT is modulated by linguistic factors, such as place of articulation (Cho and Ladefog 1999), word position (Antoniou et al. 2010), lexical stress (Casillas et al. 2015), and speech rate (Magloire and Green 1999) in monolingual and bilingual speech. For instance, faster speech is associated with shorter VOT and slower speech is associated with longer VOT, though the size of the effect may be language specific. Unfortunately, the large majority of studies related to stop contrasts in bilinguals do not take speech rate into account. One method to do so, proposed by (Stölten et al. 2015), is to use a relative measure of VOT by calculating the proportion of the duration VOT occupies in the stop + vowel sequence. This measure, relative VOT, has proven to be a more accurate, fine-grained metric that controls for possible speech rate confounds and provides more precise between-group comparisons.

In addition, bilingual language modes (Grosjean 2001) represent another factor that must be considered. Bilingual production/perception can vary (1) according to the mode (unilingual, bilingual) in which it is tested (e.g., Antoniou et al. 2010; Gonzales and Lotto 2013), and (2) as a function of the expectations the bilingual has about the communicative context (e.g., Gonzales et al. 2019; Lozano-Argüelles et al. 2020; Yazawa et al. 2019). These facts underscore the need to take special care when designing experiments so as to avoid confounds related to language modes.

Further methodological concerns include statistical power and sample size. Most studies in the social sciences test for small effects (Ellis 2010; cf. Plonsky and Oswald 2014), include small samples sizes, and, therefore, are underpowered (see Brysbaert 2020). The importance of this fact should not be overlooked, as an underpowered study is more likely to commit a type II error (false negative), and contributes to a literature with lower positive predictive value. Consequently, this implies that the prevalence of significant findings related to "compromise" categories may be indicative of publication bias.

Finally, advances in computational power have also led to more robust analytic techniques at the disposal of the speech researcher. Consider, for instance, multilevel modeling. These models provide two clear advantages: (1) they allow for partially pooled parameter estimates that are less affected by influential data points, and (2) they obviate the need to pool over subject and item repetitions in repeated measures designs. The combination of (1) and (2) helps to evade pseudoreplication in the phonetic sciences and reduces the likelihood of committing type II errors (Winter 2011).

### 1.3. The Present Study

The body of evidence suggesting that "compromise" categories in bilingual stop pro-duction may be fraught with confounds related to the primary outcome measure, as well as methodological issues related to language modes, power, sample size, and analytic techniques. This raises the question as to whether or not variable bilingual stop produc-tion may be best accounted for by some other phenomenon, such as dynamic phonetic interactions associated with language activation, rather than "compromised" underlying

representations. All of the above motivate the need to assess the cumulative evidence via meta-analytic techniques. The present study aimed to address this need. Meta-analysis offers a principled method for assessing a body of research by using independent observations to derive an average effect size and, thus, draw an overall conclusion regarding the direction and magnitude of real-world effects. The "compromise" category literature would particularly benefit from meta-analysis because a large amount of research and theory building has been based on the early findings.

In order to assess the "compromise" category literature, the present study addressed the following questions:

1. What is the cumulative evidence for "compromise" categories?
2. Is the effect modulated by linguistic factors, such as place of articulation, word position, or lexical stress?
3. Do analytic techniques account for between-study variability?
4. Is the "compromise" category literature sufficiently powered?

In what follows, the present project responds to the aforementioned research questions by presenting a meta-analysis of the "compromise" VOT literature. Subsequently, this study provides an alternative account to the notion of "compromise" categories in early bilinguals using data from coronal stops.

## 2. Meta-Analysis

### 2.1. Method

#### 2.1.1. Study Identification and Screening

The analysis employed a variety of techniques to locate relevant primary studies, focusing first on amassing a large study pool and later filtering out redundant or unusable records. The first step included searching library-housed online databases using various combinations of relevant keywords. The terms 'compromise categories', 'merged categories', 'mixed categories', and 'intermediate categories' were searched individually and in combination with 'early learners', 'early bilinguals' or 'simultaneous bilinguals', as well as 'VOT' or 'voice-onset time'. The databases included ERIC, Science Direct, Linguistics and Language Behavior Abstracts, PsycINFO, ProQuest Dissertations and Theses, and FirstSearch (see the supplementary materials for more details on the results from each search). Ancestry studies and studies citing the primary studies were also obtained via searches in Google and Google scholar. When potential studies were not available through the aforementioned resources, authors were contacted directly. There were 153,860 re-cords identified through database searching and 27 additional ancestry studies identified through Google and Google scholar. After removing duplicates and irrelevant hits, the study pool contained 148 records.

#### 2.1.2. Eligibility

The 148 full-text articles and dissertations were assessed for eligibility. To be eligible, a study had to (1) include simultaneous and/or early bilinguals (AOA before 12 years old) that were adults at the time of testing, (2) examine a language pair with a two-way stop voicing contrast, specifically voiceless stops, and (3) include a monolingual comparison group.[2] For all studies, data from both languages were included in the meta-analysis if the participants were simultaneous bilinguals and control comparisons were included. For sequential learners, their second, sequentially learned language was utilized in comparison with controls, as "compromise" phonetic categories are more common in the literature in the L2. Languages with three-way contrasts (namely Korean) were excluded because these contrasts typically involve other parameters, i.e., pitch (see Holliday 2015). Finally, the study pool was limited to analyses of voiceless stops due to the fact that the majority

---

[2] An anonymous reviewer duly notes the fact that there is currently debate surrounding the use of monolingual control populations in bilingual research (see Sakai 2018). The present meta-analysis included monolingual controls as part of the search criteria because it best reflects the practices of research on "compromise" categories and thus led to the largest number of potential studies for the dataset.

of the records identified involved English, which allows both short-lag and lead VOT for phonologically voiced stops.

The assessment resulted in a dataset of 68 studies that appeared to meet the aforementioned inclusion criteria. The studies spanned 6 decades, from the 1970s to the present, and came from a variety of sources, including journal articles, book chapters, MA and PhD theses, conference proceedings, and unpublished manuscripts. Forty-eight studies were discarded for a number of reasons: (1) they looked at something different (k = 15), (2) they did not include a control group for comparison (k = 16), (3) there was missing data (k = 10), (4) this study included duplicate data presented in a prior study (k = 2), or (4) this study examined a three-way contrast (k = 5). Requests for missing or unreported data were sent via email. One response out of 10 requests was received (with data). The search process led to a final dataset comprising 20 studies with 37 independent comparisons and a pooled participant sample size of 641.The average age cut-off used to classify participants as early bilinguals was 4.53 ± 2.84 SD. The majority of the usable studies were journal articles (n = 16), followed by MA/PhD theses and conference proceedings (n = 4). The usable studies spanned 5 decades, with 3 from the 1980s, 4 from the 1990s, 4 from the 2000s, 8 from the 2010s, and 1 study from 2020.

### 2.1.3. Coding

Each study was coded for linguistic and methodological features and effect sizes. The effect size was a measure of standardized mean difference (SMD), specifically Hedge's g. The linguistic features included the stop category (/p/, /t/, or /k/), lexical stress (stressed or unstressed syllable), and word position (initial, medial). The methodological features included analytic strategy (*t*-test, ANOVA, or LME) and pooling method for stops (individual evaluations of each segment versus averaging over combinations of segments). Effect sizes were calculated primarily using reported means and standard deviations (or standard errors).[3] Ultimately, word position was excluded as a moderator, as there were not enough studies that included this factor.

### 2.1.4. Statistical Analysis

A cross-classified Bayesian meta-analysis was conducted by fitting the study data with the multilevel regression model formulated below:

$$\begin{aligned}
\text{SMD}_i &\sim \text{Normal}(\theta_i, \sigma_i = \text{se}_i) \\
\theta_i &\sim \text{Normal}(\mu, \tau) \\
\mu &\sim \text{Normal}(0, 1) \\
\tau &\sim \text{HalfCauchy}(0, 1)
\end{aligned} \tag{1}$$

Effect size (SMD) was the outcome variable and lexical stress (stressed, unstressed) and analytic strategy (LME, other) were included as population-level effects (i.e., fixed effects). The likelihood of the outcome variable was assumed to be a gaussian distribution. Individual studies and stop pooling methods were group-level effects (i.e., random effects). Population-level effects were deviation coded (lexical stress: stressed = 0.5, unstressed = −0.5; analytic strategy: LME = 0.5, other = −0.5) such that the posterior distribution of model estimates for each effect provided an assessment of effect size. The model included regularizing, weakly informative priors (Gelman et al. 2017) which were normally distributed and centered at 0 with a standard deviation of 1 for all population-level parameters. A cauchy prior set at 0 with scale 1 was used for $\tau$. Fianlly, the model was fit with 4000 iterations (2000 warm-up) and Hamiltonian Monte-Carlo sampling was carried out with

---

[3]　In the case of one study, effect size was calculated from the reported degrees of freedom and F-value of a one-way ANOVA. For three studies standard deviations were not reported but the manuscript included boxplots. In these cases, the median and interquartile range were derived from the boxplots via webplot digitizer (Rohatgi 2020) and then used to calculate the mean and standard deviations (see Wan et al. 2014). All figures used for these approximations are available in the supplementary materials.

4 chains distributed across 4 processing cores. The analysis was conducted in R (R Core Team 2019, version 4.0.3) and was fit using stan (Stan Development Team 2018) via the R package brms (Bürkner 2017). More information regarding Bayesian Data Analysis is available in the supplementary materials. All supplementary analyses as well as the data, code, and the experimental materials necessary to reproduce the analyses reported in this article are available at: https://osf.io/un45x/.

### 2.2. Results

Summaries of the posterior distribution of the meta-analytic model are provided in Figures 1 and 2 (see supplementary materials for summaries in table format). Averaging over lexical stress and analytic strategies, the pooled estimate of the standardized mean difference (SMD) was small and negative ($\beta = -0.132$, HDI = [$-0.708$, 0.468]). As reflected in Figure 1A, the posterior distribution of the effect size estimate is wide and encompasses plausible values on both sides of a point null of 0. In short, there is not compelling evidence in support of a difference in VOT between bilinguals and monolingual controls. The estimates of variability from group-level effects *pooling method* and *individual studies* are plotted in Figure 1B. One can see that *individual studies* were a considerable source of variability. Figures 1C and 2 illustrate estimate uncertainty in comparison with the overall pooled effect. The moderator (subgroup) effects were negligible. Specifically, lexical stress had no effect on group differences ($\beta = -0.14$, HDI = [$-0.766$, 0.483]), nor did analytic strategy ($\beta = 0.182$, HDI = [$-0.657$, 1.035]), though mixed effects models tended to narrow the gap between group difference estimates, as illustrated by the positive $\beta$-parameter (see Figure 1D).

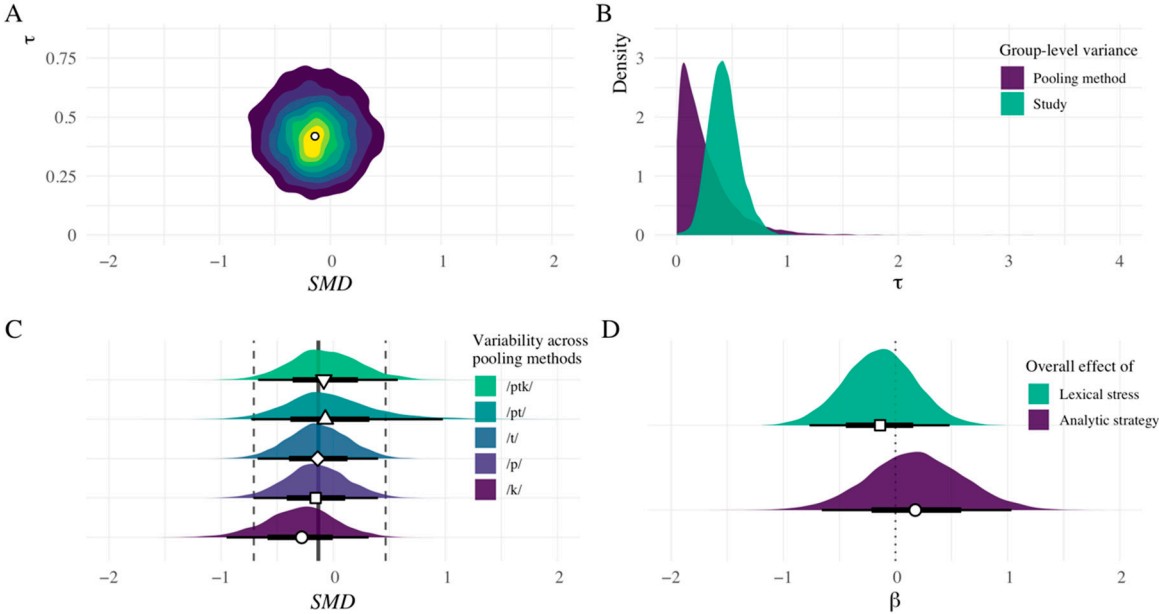

**Figure 1.** Summary of the posterior distribution of the meta-analytic model. Panel (**A**) plots the pooled estimate of effect size (SMD) on the horizontal axis and standard error ($\tau$) on the vertical axis. Lighter colors illustrate higher density areas (i.e., more plausible values) and the point represents the posterior median. Panel (**B**) illustrates estimates of variance from group-level effects and Panel (**C**) provides a sub-category summary of variability as a function of pooling method. The vertical lines show the posterior median (solid) and 95% credible interval (dashed) of the pooled effect. Panel (**D**) summarizes the posterior distributions of the overall effects of the lexical stress and analytic strategy moderators.

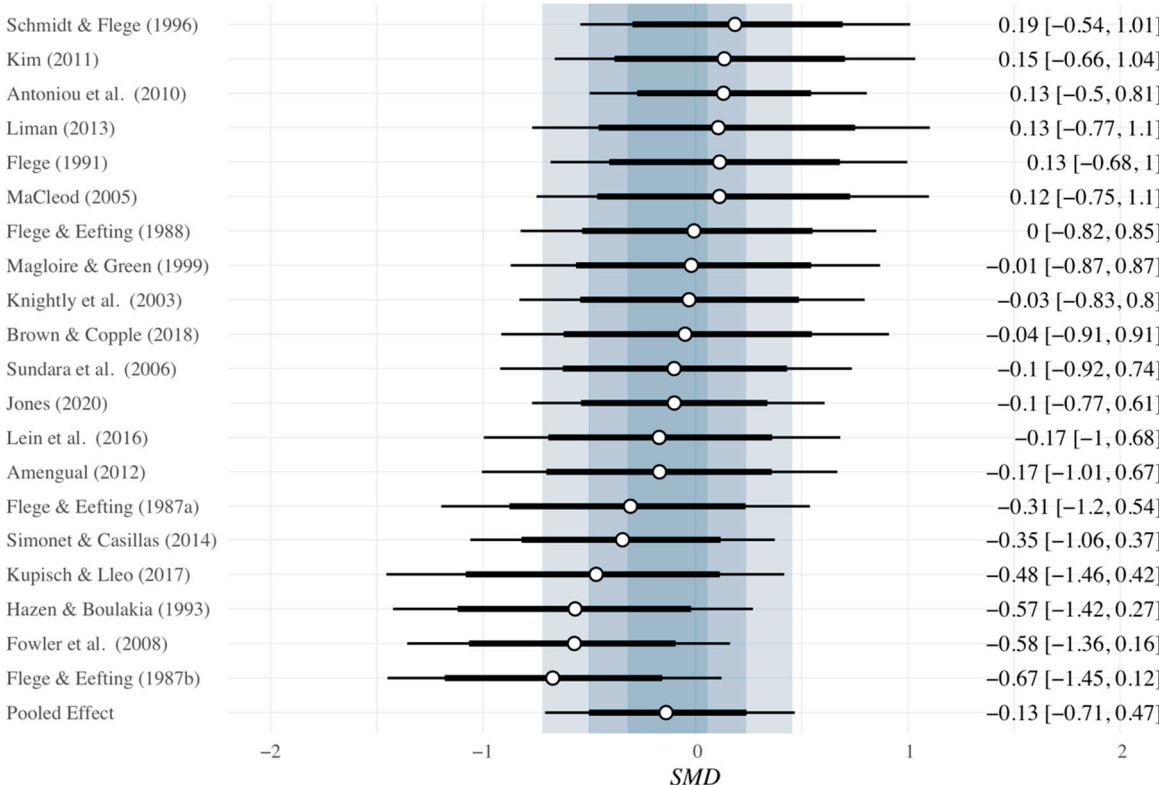

**Figure 2.** Summary of posterior model estimates for individual studies (vertical axis) and the effect size (SMD, horizontal axis). White points represent posterior medians and horizontal bars capture the ± 95% and 80% credible intervals, which are also printed along the right vertical margin. The vertical shaded rectangle illustrates ± 95%, 80%, and 50% credible intervals around the posterior median of the pooled effect.

### 2.3. Interim Discussion

Using a variety of techniques to locate relevant research on bilingual stop production, a dataset of 20 studies with 37 independent, usable bilingual-to-monolingual comparisons was collected. The studies were coded for linguistic and methodological features and effect sizes were calculated in order to estimate the cumulative effect in the literature comparing bilingual voiceless stop production to that of monolingual controls via meta-analysis. The results of the meta-analysis suggest that the cumulative effect in the literature is negligible and includes a high degree of uncertainty. The pooled estimate of the present dataset is −0.132 standard deviations, 95% HDI = [−0.708, 0.468]. Traditional standards classify an effect size of 0.20 as small, 0.50 as medium, and 0.80 as large (see Cohen 2013; Ellis 2010). Plonsky and Oswald (2014) suggest even more stringent standards for L2 research (small: d = 0.40, medium: d = 0.70, large: d = 1.00). In this dataset, the posterior probability that the average effect size meets or exceeds Plonsky and Oswald (2014) suggestion for a small effect is 0.18. Why, then, has a significant amount of the literature built theory around the "compromise" category claim?

One possibility is publication bias. Figure 3 provides a funnel plot of the unpooled dataset comparing standard error as a function of effect size (SMD). If the literature on bilingual stop production suffered from publication bias one would expect to see individual studies (points) dispersed asymmetrically around the pooled estimate. This does not appear to be the case, as one observes studies on both sides of the pooled estimate (the vertical black line).

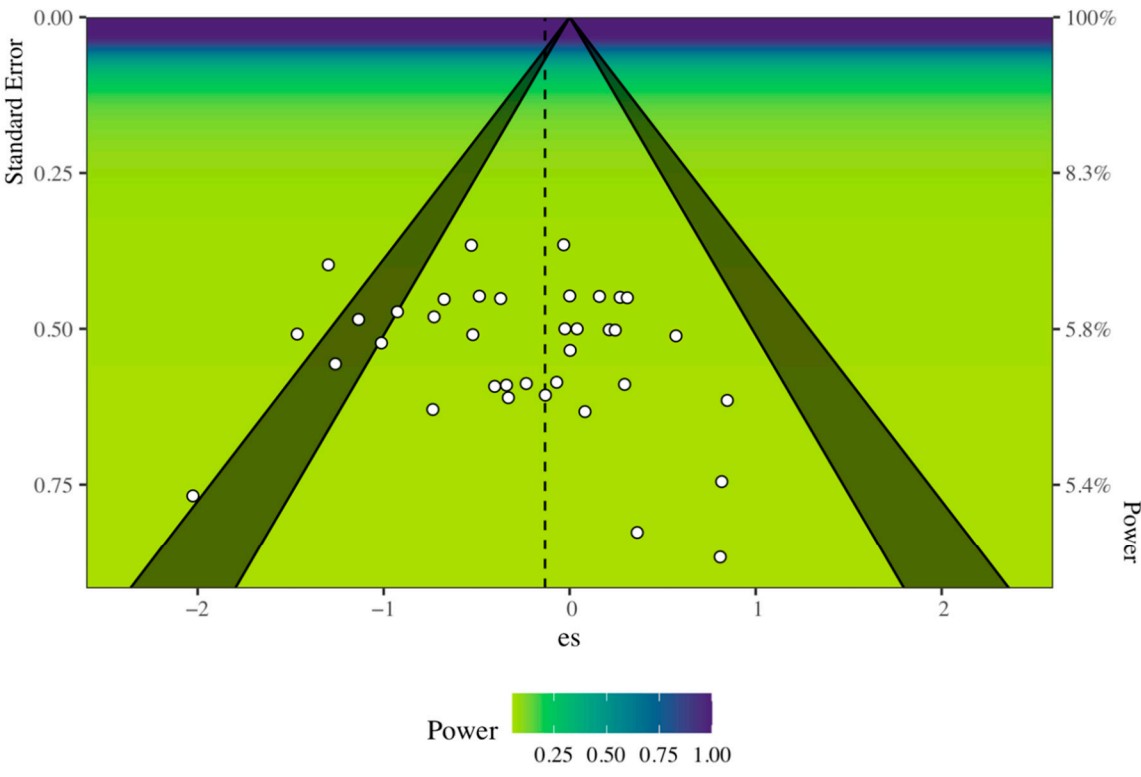

$\alpha = 0.05, \delta = -0.13 \mid \text{med}_{\text{power}} = 5.8\%, d_{33\%} = 0.77, d_{66\%} = 1.21 \mid E = 2.13, O = 6, p_{\text{TES}} = 0.006, \text{R-Index} = 0\%$

**Figure 3.** Funnel plot of effect sizes (Hedge's g, horizontal axis), standard error, and power (vertical axis). Power for individual studies was calculated using the pooled estimate ($-0.132$ SMD, vertical dashed line) from the meta-analytic model. Background color illustrates power (darker colors represent more power).

If we were to assume that the "compromise" category effect in bilingual speech were real, then we are left questioning why it was not borne out in the meta-analysis. One possibility, a glaring shortcoming of the extant literature, is the fact that all of the studies analyzed are underpowered (median = 5.80%). The gold standard for power suggested in psychology is 80% (see Ellis 2010), and none of the studies included in the present analysis met this standard (see right margin of Figure 3). The lack of power in this literature likely results from small sample sizes. If we again assume that "compromise" categories exist in bilingual speech and that the effect is small—0.40 using Plonsky and Oswald's suggestion for L2 research—a hypothetical study would need to include 99 participants *per group* to have an 80% chance of capturing the effect with alpha set a 0.05. The number is even more astounding if we use the pooled estimate of the present study and assume the same conditions: 896 participants, again, *per group*.

Another possible factor contributing to the low overall power in the extant literature is measurement error. While measuring VOT is rather straightforward, the majority of the early research in this area required researchers to average VOT values for individuals over items, as well as over repetitions of items. This suggests that the values being used in statistical analyses may inherently misrepresent the actual productions of the participants. If, for example, a participant produces the voiced stops of English variably, with and without pre-voicing, a mean value for this participant might not accurately represent a prototypical stop from either distribution. Given the structured variability observed in monolingual (Chodroff and Wilson 2017) and bilingual (Chodroff and Baese-Berk 2019) stop production, it seems reasonable to assume that there is also inherent structure manifested in other ways. Advancing technology has made the costly computations involved with partial pooling methods more accessible to researchers, and, as a result, this is reflected in more recent studies using more powerful analytic strategies, such as multilevel models. The present meta-analysis took analytic strategies into account and found no effect, though

this may be explained by the fact that the majority of the studies included did not use partial pooling (k = 33 out of 37 independent comparisons).

It is important to note that the meta-analysis presented here is limited in several non-trivial ways. First, the pool of studies excluded clearly relevant, seminal research due to uncontrollable circumstances (e.g., missing data). Second, the criteria for inclusion necessarily limited the sample to voiceless stop production from early/simultaneous bilinguals speaking languages with two-way contrasts. While this filtering facilitated controlling possibly confounding factors, it raises questions regarding what data from other sources can contribute to the cumulative research on "compromise" categories in bilinguals. If, for example, "compromise" categories are not real in early and simultaneous bilinguals, what, then, can explain the observed variability in bilingual stop production? The following section considers coronal stop data from highly proficient Spanish–English early bilinguals in order to explore the interaction between language-specific voice-timing and place of articulation differences when both languages are activated in bilingual mode.

## 3. Production of Coronal Stops

This section presents an acoustic analysis of coronal stop data from early Spanish–English bilinguals in order to explore alternative explanations for "compromise" categories. The bilinguals completed a delayed shadowing task in which both languages were highly activated in order to elicit bilingual language mode. While English contrasts /d t/ through short-lag VOT and long-lag VOT, respectively, Spanish has the same voicing distinction through lead VOT and short-lag VOT. Importantly, the coronal stops of each language also differ regarding place of articulation. In Spanish coronal stops are described as dental, whereas in English they are described as alveolar (Casillas et al. 2015).

### 3.1. Method

#### 3.1.1. Participants

The dataset included 33 participants between the ages of 18 and 23, all of which were female. There were 17 bilingual participants and 16 monolingual controls. Of the monolingual participants, eight were native Spanish speakers, born and raised on the island of Majorca, Spain. The remaining eight were native English speakers, born and raised in the US Southwest.

The Spanish–English bilinguals came from Southern Arizona and Northern Mexico. They were raised in Spanish-speaking families and were schooled mostly in English in Southern Arizona. They reported using English and Spanish daily, both in the classroom as well as with their friends and relatives. The bilingual group completed the Bilingual Language Profile (BLP, Gertken et al. 2014) in order to assess language dominance. The BLP calculates a weighted average of language dominance based on the individual history, use, proficiency, and attitudes of the bilinguals with regard to their languages. The measure ranges from −218 to 218 with values near the extremes implying dominance in one of the languages. Values close to 0 are taken as an indication of balanced bilingualism. In the present study, Spanish was arbitrarily assigned to positive values. Figure 4 plots language dominance (Panel A) and language use and proficiency data (Panel B) derived from the BLP. The bilingual group had a mean dominance score of 2.08 (SD = 40.42), suggesting rather balanced bilingualism (Panel A). Participants that reported using Spanish more often also tended to report being more proficient in that language; the converse was also true for English (Panel B).

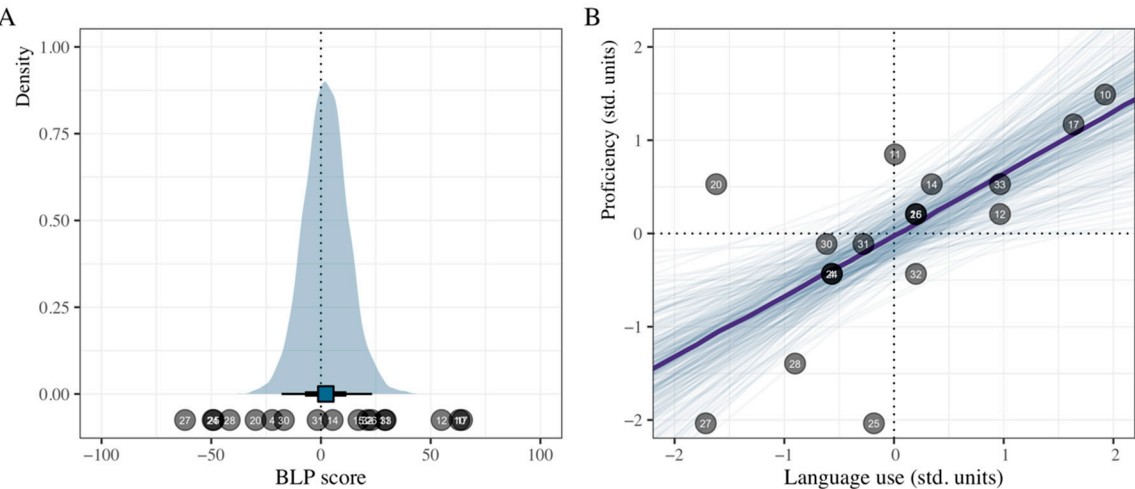

**Figure 4.** Bilingual Language Profile data. Panel (**A**) illustrates the distribution of BLP dominance scores. Panel (**B**) shows language proficiency as a function of language use. Both measures come from self-report data and are plotted in standardized units.

### 3.1.2. Materials

There were 48 target words (English: k = 24; Spanish: k = 24) that contained voiced and voiceless coronal stops in word initial position. For each language, 12 targets began with /d/ and 12 began with /t/, equally divided between stressed and unstressed syllables (see supplementary materials). All stops were followed by a low vowel (/a/ for Spanish and /æ, ɑ/ for English).

The participants completed a delayed repetition task in which they heard the target words presented in a carrier phrase ("x is the word" or the Spanish equivalent "x es la palabra"). The auditory stimuli were recordings of six male native speakers: three native English speakers and three native Spanish speakers. These recordings served as the auditory stimuli repeated out loud by the participants in the delayed repetition task. Words not containing coronal stops were considered distractors (k = 20). Praat was used to present the sentences randomly in auditory form and the speakers were asked to listen to the entire sentence and then repeat it out loud at their own pace.

The monolingual English speakers and bilinguals were recorded in a sound attenuated booth. The monolingual Spanish speakers were recorded in a quiet classroom on the campus of the *Universitat de les Illes Balears* in Majorca, Spain. The monolingual English speakers were recorded in English and the monolingual Spanish speakers were recorded in Spanish. The Spanish–English bilinguals were recorded in both of their languages in a single session with all English and Spanish items presented in a single, randomized block in order to activate both languages. The full dataset included 3519 tokens (24 target words per language × 3 repetitions). Eighty-one items (2.25%) were discarded due to mispronunciations or extraneous noise. A Shure SM10A dynamic head-mounted microphone with a Sound Devices MM-1 microphone pre-amplifier captured the acoustic signal and it was saved to a Marantz PMD660 digital speech recorder. The signal was digitized at 44.1 kHz and 16-bit quantization.

### 3.1.3. Measurements

The audio files were low-pass filtered at 11.025 kHz. Synchronized waveform and spectrographic displays were used to mark the onset of modal voicing and of the stop burst, as well as the offset of the first vowel. The onset of voicing was operationalized as the upwards zero-crossing of the first periodic pattern in the oscillogram and the offset of the vowel was marked at the downwards zero-crossing of the final periodic pattern. VOT was calculated as the difference (in ms) between the onset of modal voicing and the onset of the burst. Relative VOT was calculated as the ratio between VOT and the total

duration of the stop-vowel sequence. Spectral moment measures were calculated from a 6 ms window beginning at the onset of the burst. Specifically, kurtosis was extracted from the spectral envelope, which ranged from 60 Hz to 11.025 kHz.

### 3.1.4. Statistical Analysis

The bilingual voice-timing data were analyzed using Bayesian multilevel regression models. Specifically, separate models were fit for VOT and relative VOT as a function of voicing (voiced, voiceless) and language (English, Spanish). Fixed effects were deviation coded (voicing: voiced = 0.5, voiceless = −0.5; language: English = 0.5, Spanish = −0.5) thus the posterior distribution provided an assessment of effect size for each predictor. Item repetition was included as a continuous predictor and was centered to have a mean of 0. The random effects structure included a by-subject intercept with a random slope for voicing, as well as a by-item intercept. The model included regularizing, weakly informative priors (Gelman et al. 2017) which were normally distributed and centered at 0 for all population level parameters. The standard deviation was set at 50 and 3 for the VOT and relative VOT models, respectively. The models were fit with 4000 iterations (1000 warm-up) and Hamiltonian Monte-Carlo sampling was carried out with 4 chains distributed across 4 processing cores. For each model a region of practical equivalence (ROPE) was established around a point null value of 0 (see Kruschke 2018) using the following formula:

$$\frac{\mu_1 - \mu_2}{\sqrt{\frac{\sigma_1^2 + \sigma_2^2}{2}}} \tag{2}$$

For all models, median posterior point estimates are reported for each parameter of interest, along with the 95% highest density interval (HDI), the percent of the region of the HDI contained within the ROPE, and the maximum probability of effect (MPE). For statistical inferences, a posterior distribution for a parameter β in which 95% of the HDI falls outside the ROPE and a high MPE (i.e., values close to 1) were taken as compelling evidence for a given effect. Again, the analyses were conducted in R (R Core Team 2019, version 4.0.3) and models were fit using stan (Stan Development Team 2018) via the R package brms (Bürkner 2017).

### 3.2. Results

Figure 5 plots the VOT data as a function of group and stop category. Looking across the horizontal axis, one can observe distinct distributions for voiced and voiceless stops. In comparing the three groups along the vertically faceted panels, it becomes particularly clear that the bilinguals produce the coronal stops similarly to the monolingual controls in English and Spanish; however, upon close inspection, we can see that the bilingual group produces more pre-voiced /d/ tokens in English than the monolingual English speakers, as well as more short-lag /d/ tokens than monolingual Spanish speakers. When pooled together to calculate by-subject averages, tokens such as these could skew the measurement, fostering the notion that the bilinguals produce some segments with intermediate values that do not correspond with prototypical monolingual values of either language. The analysis of the voice-timing data did not provide compelling evidence that the bilingual productions differed from monolingual controls.

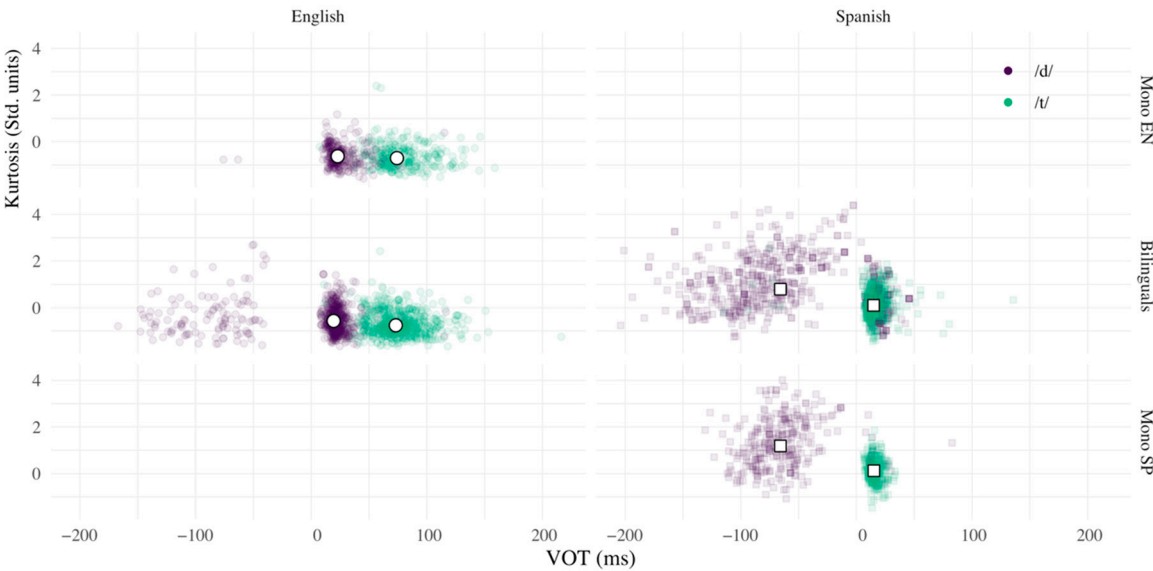

**Figure 5.** Voice-onset time (horizontal axis) and Kurtosis (vertical axis) as a function of speaker group and stop category. Colored points represent raw data. White points represent category medians.

### 3.2.1. VOT

For Spanish, phonologically voiced stops had shorter VOT than voiceless stops ($\beta$ = −38.865, HDI = [−46.555, −31.766], ROPE = 0, MPE = 1), but VOT did not vary as a function of group ($\beta$ = 0.78, HDI = [−6.977, 8.604], ROPE = 0.844, MPE = 0.579), nor did the two factors interact ($\beta$ = 1.213, HDI = [−4.842, 8.058], ROPE = 0.898, MPE = 0.646). For English stops, voiced stops also had shorter VOT than voiceless stops ($\beta$ = −31.743, HDI = [−37.77, −25.363], ROPE = 0, MPE = 1). Averaging over stop category, the analysis suggested that bilingual VOT was slightly shorter than that of the monolingual English speakers ($\beta$ = −7.005, HDI = [−13.015, −0.951], ROPE = 0.19, MPE = 0.988), though approximately 19% of the most plausible estimates fell within our *a priori* established region of practical equivalence. There was no evidence of a group × phoneme interaction ($\beta$ = −5.161, HDI = [−11.02, 0.18], ROPE = 0.41, MPE = 0.966).

### 3.2.2. Relative VOT

The more stringent relative VOT metric tells a similar story. For Spanish, voiced stops comprised a larger proportion of the stop-vowel sequence than voiceless stops ($\beta$ = 0.1, HDI = [0.071, 0.127], ROPE = 0, MPE = 1), but the stop-vowel ratio did not vary between groups ($\beta$ = −0.001, HDI = [−0.018, 0.015], ROPE = 0.945, MPE = 0.576), nor was there a two-way interaction ($\beta$ = −0.004, HDI = [−0.028, 0.017], ROPE = 0.792, MPE = 0.658). For English, voiced stops comprised a smaller proportion of the stop-vowel sequence than voiceless stops did ($\beta$ = −0.101, HDI = [−0.119, −0.083], ROPE = 0, MPE = 1). Averaging over stop category, there was no difference between groups ($\beta$ = 0.004, HDI = [−0.013, 0.02], ROPE = 0.898, MPE = 0.697), and, finally, the two factors did not interact ($\beta$ = 0.01, HDI = [−0.004, 0.025], ROPE = 0.699, MPE = 0.911).

The complete summary of the posterior distributions of all models are available in Table 1. As a point of comparison, the VOT and relative VOT data were also fit using a 2 × 2 repeated measures ANOVA under a null-hypothesis significance testing (NHST) frequentist framework by averaging over items and item repetitions. The VOT model, and not the relative VOT model, suggested a group × stop category interaction for the English data. The complete model summaries for all analyses are available in the supplementary materials.

**Table 1.** Summary of the posterior distribution modeling voiceless responses as a function of VOT, context, z-LexTALE, and order. The table includes posterior means, the 95% HDI, the percentage of the HDI within the ROPE, and the maximum probability of effect (MPE).

| Outcome | Language | Parameter | β | 95% HDI | MPE | ROPE % | ROPE |
|---|---|---|---|---|---|---|---|
| VOT | Spanish | Intercept | −23.19 | [−31.84, −14.89] | 1 | 0 | [−5.12, 5.12] |
| | | Group | 0.78 | [−6.98, 8.60] | 0.58 | 0.84 | [−5.12, 5.12] |
| | | Phon. | −38.87 | [−46.56, −31.77] | 1 | 0 | [−5.12, 5.12] |
| | | Item rep. | −1.02 | [−2.73, 0.68] | 0.89 | 1 | [−5.12, 5.12] |
| | | Group × Phon. | 1.21 | [−4.84, 8.06] | 0.65 | 0.89 | [−5.12, 5.12] |
| | English | Intercept | 45.19 | [38.66, 51.81] | 1 | 0 | [−4.60, 4.60] |
| | | Group | −7.00 | [−13.02, −0.95] | 0.98 | 0.19 | [−4.60, 4.60] |
| | | Phon. | −31.74 | [−37.77, −25.36] | 1 | 0 | [−4.60, 4.60] |
| | | Item rep. | −0.21 | [−1.57, 1.13] | 0.62 | 1 | [−4.60, 4.60] |
| | | Group × Phon. | −5.16 | [−11.02, 0.18] | 0.96 | 0.41 | [−4.60, 4.60] |
| Relative VOT | Spanish | Intercept | 0.21 | [0.19, 0.23] | 1 | 0 | [−0.01, 0.01] |
| | | Group | −0.01 | [−0.02, 0.02] | 0.58 | 0.94 | [−0.01, 0.01] |
| | | Phon. | 0.10 | [0.07, 0.13] | 1 | 0 | [−0.01, 0.01] |
| | | Item rep. | 0.01 | [−0.01, 0.01] | 0.83 | 1 | [−0.01, 0.01] |
| | | Group × Phon. | −0.01 | [−0.03, 0.02] | 0.66 | 0.79 | [−0.01, 0.01] |
| | English | Intercept | 0.29 | [0.27, 0.31] | 1 | 0 | [−0.01, 0.01] |
| | | Group | 0.01 | [−0.01, 0.02] | 0.69 | 0.89 | [−0.01, 0.01] |
| | | Phon. | −0.10 | [−0.12, −0.08] | 1 | 0 | [−0.01, 0.01] |
| | | Item rep. | −0.01 | [−0.01, 0.01] | 0.83 | 1 | [−0.01, 0.01] |
| | | Group × Phon. | 0.01 | [−0.01, 0.03] | 0.91 | 0.69 | [−0.01, 0.01] |

### 3.2.3. Bilingual Performance Mismatches

Taken together, the aforementioned analyses do not provide compelling evidence that Spanish–English bilinguals produce Spanish and English coronal stops in a manner that robustly differs from their monolingual counterparts. That being said, there does appear to be a qualitative difference between the bilinguals and the monolingual controls in terms of variability. Specifically, on occasion, the bilinguals appear to produce English /d/ with pre-voicing, as well as Spanish /d/ and English /t/ with short-lag VOT. In order to analyze this further, the data were subset based on these mismatched VOT properties and the relationship between voice timing (relative VOT) and place of articulation (Kurtosis of the stop burst) was explored.

The scatter plots in Figure 6 show the mismatched targets for /d/ (Panel A) and /t/ (Panel B) as a function of target language. The vertical axis is kurtosis (standardized units) and the horizontal axis is relative VOT (standardized units). Higher kurtosis values are found in Spanish monolingual coronals with regard to English monolingual coronals. This difference reflects the place of articulation differences between Spanish (dental) and English (alveolar) coronals (see Casillas et al. 2015; Sundara et al. 2006).

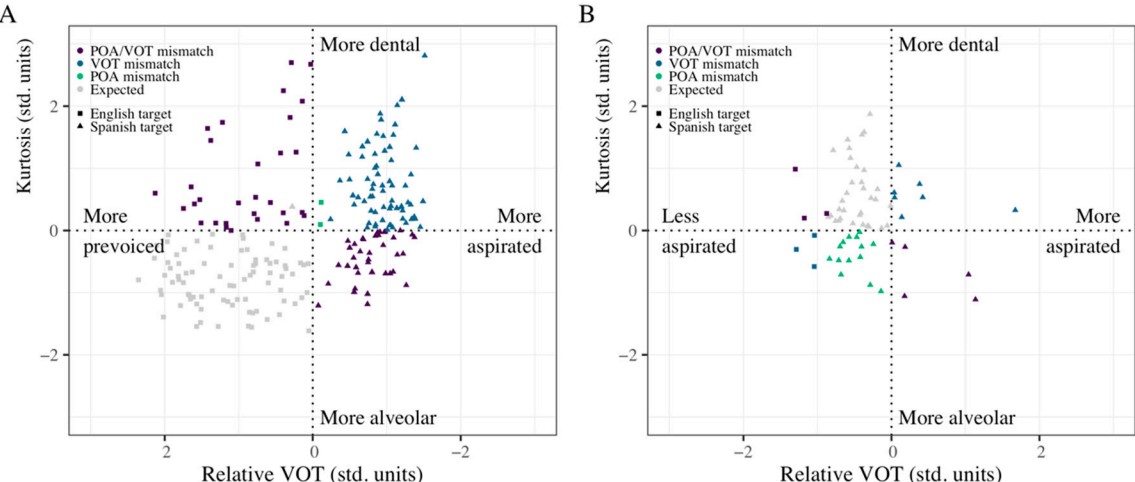

**Figure 6.** Summary of performance mismatches for /d/ targets (Panel **A**) and /t/ targets (Panel **B**) in bilinguals. In both panels, English targets are triangles and Spanish targets are squares. The horizontal axis represents relative VOT and the vertical axis represents kurtosis in standardized units.

The plots can be interpreted using the quadrants specified by the vertical and horizontal dotted lines. Points on the left side of the vertical line are associated with lower VOT values and points to the right side are associated with higher VOT values.[4] Points above the horizontal dotted line contain burst characteristics consistent with dental place of articulation (POA), while points below the horizontal line contain burst characteristics consistent with alveolar POA. The colors of the points are associated with the type of mismatch, which could be VOT, POA, or both VOT and POA (grey points are tokens produced as expected that met the filtering criteria). Of particular interest are the VOT/POA mismatches in the upper-left and lower-right quadrants of both plots.

The majority of the mismatched productions occurred in phonologically voiced target items (Panel A). One can observe Spanish targets that were realized as short-lag stops with more alveolar bursts (purple triangles), as well as English items that were produced with pre-voicing and more dental bursts (purple squares). The phonologically voiceless target items led to fewer mismatches (Panel B), though there are instances of English targets with short-lag VOT and more dental bursts, as well as Spanish targets with long-lag VOT and more alveolar bursts. The amount of mismatches produced was not associated with language dominance or any of the self-report measures collected in the BLP (see supplementary materials for more details).

## 4. Discussion

The present study included two primary analyses: a meta-analysis of "compromise" VOT and coronal stop data from a delayed shadowing production experiment. The results of the meta-analysis suggested that the pooled estimate of the cumulative effect-size was small, and just as likely to be positive as it was to be negative. The model considered linguistic factors as well as methodological factors. There was no evidence that lexical stress was a relevant moderator for "compromise" VOT, nor that analytic strategies resulted in a higher or lower likelihood of encountering differences between bilinguals and monolinguals. Between-study differences accounted for a larger proportion of the variance than pooling methods for the stop categories. The posterior estimate of the pooled effect made it particularly clear that the "compromise" VOT literature is underpowered (median = 5.80%).

---

[4]  Note that the *x* axis was reversed in panel A of Figure 6 so that both panels are interpreted in the same manner, that is, with lower VOT values on the left and higher VOT values on the right. The reason the *x* axis must be reversed for this to be true is because pre-voiced stops have higher relative VOT values, i.e., they account for a larger portion of the stop + vowel sequence, than short-lag stops. The same is true for long-lag stops when compared with short-lag stops.

That being said, there is no evidence suggesting that this literature suffers from publication bias.

The analysis of the coronal stop data produced two findings. First, bilingual stop production is highly variable, and, second, this variability is structured in consistent, predictable ways. Specifically, the analysis showed that the bilingual speech contained target mismatches. That is, when producing English targets, the bilingual speech included pre-voiced /d/ tokens and short-lag /t/ tokens. When speaking Spanish, the bilingual speech included short-lag /d/ tokens. Crucially, these mismatches also displayed burst characteristics that were consistent with the mismatched language, suggesting that, on the whole, not only were the bilinguals using the voice timing of their other language, but also the corresponding place of articulation. The performance mismatches are attributed to the fact that the experiment induced high activation of both Spanish and English by being conducted in "bilingual" mode.

Taken together, the results of the present work depict bilingual stop production in a new light. Specifically, it seems unlikely that early bilinguals have "compromise" categories for stops, but rather produce performance category mismatches that result from dynamic phonetic interactions associated with language activation. This assertion is inconsistent with the evidence put forth by many studies in this literature. Possible explanations for the discrepancy may revolve around methodological issues. For example, the effects of bilingual language modes on speech production/perception are well attested. Many, but not all, of the earliest studies in the "compromise" VOT literature control for language mode by conducting experimental sessions on different days, with all materials and interactions with the experimenter conducted in the relevant language. Thus, it seems unlikely that language mode is a confounding factor in this body of literature. A more likely candidate is speech rate, which is negatively correlated with VOT (see Schmidt and Flege 1996; Magloire and Green 1999). Stölten et al. (2015) proposed controlling for speech rate by using relative VOT when making between group comparisons. Only two of the studies included in the present analysis controlled for speech rate. Magloire and Green (1999) found that Spanish–English bilinguals' English bilabial stop production was no different from that of monolingual English speakers in slow, normal and fast speech.[5]

The SLM posits that if learners can avoid the equivalence classification of two similar phones, they will be able to develop new phonetic categories. When this occurs, one prediction is that the L2 phonetic category will deflect or dissimilate from neighboring categories in order to maintain phonetic contrast. The early bilinguals of the present study clearly established a phonetic category for English /t/. In terms of the equivalence classification, it is more difficult to make a determination regarding /d/, which was realized with pre-voicing at a higher rate than in the monolingual controls. Any claim that increased pre-voicing is indicative of a "compromise" category would be weak, at best, as American English stops are often pre-voiced (Lisker and Abramson 1964) and may even be the default for some varieties (Walker 2020). Given the acoustics of the category mismatches, there does not appear to be any reason to believe that the bilinguals underlying phonetic categories are anything different from the input they receive in their speech community. On the contrary, the results herein support models that do not assert a bi-directional influence on the underlying grammar.

The Second Language Linguistic Perception Model (L2LP, Escudero 2005; Van Leussen and Escudero 2015), for example, proposes that the L2 grammar is separate from and develops independently of the L1 grammar. This model conceives of Grosjean's bilingual language modes as a continuum ranging from a unilingual mode in the L1 on one extreme to a unilingual mode in the L2 on the other, with an L1–L2 bilingual mode in the middle. Importantly, the L1 and L2 underlying grammars can be activated selectively or in parallel in real time. Language activation can be triggered by variables that are linguistic or

---

5   Other studies on bilingual stop production have taken speech rate into account, to be sure, but were not included in the present analysis due to missing and/or unavailable data.

extralinguistic in nature, such as the use of cognates in the experimental items (e.g., Amengual 2012) or the participant's beliefs about the language required for a given task. Thus, this model can account for phenomena like the double phonemic boundary effect (e.g., Lozano-Argüelles et al. 2020) and language-dependent cue weighting (e.g., Yazawa et al. 2019) in speech perception.

To the author's best knowledge, the L2LP has not been used to account for production of consonants, but, on the surface, it appears to provide an elegant explanation for bilingual performance mismatches via parallel activation in bilingual mode. For instance, the bilinguals in the present study would have a monolingual Spanish mode on one end, where they produce /d t/ as dental stops with pre-voicing and short-lag VOT, respectively, and a monolingual English mode on the other, where /d t/ are realized at alveolar place with short- and long-lag VOT. When both languages are activated in parallel in bilingual mode, all combinations of place and voice settings are available, though not equally probable.

It is worth noting that neither the SLM nor the L2LP were designed to explain simultaneous bilingualism. Moving forward, a complete model of bilingual phonology should be able to account for behavior of both simultaneous and sequential bilinguals with a wide variety of linguistic experience in order to appropriately model the nature of the dynamic phonetic interactions that occur between robust phonetic sub-systems in diverse communicative contexts.

The present work could be improved by including more of the extant literature on "compromise" VOT. The meta-analysis presented here excluded a non-trivial subset of relevant studies due to missing data. As mentioned in the interim discussion, the whole of the "compromise" VOT literature includes small sample sizes and is underpowered. The coronal stop data presented herein is subject to the same critique. This analysis was necessarily exploratory in nature, serving the main purpose of providing a qualitative assessment of performance mismatches. Future studies should further examine the nature of performance mismatches to shed light on how they might be modulated by proficiency, language dominance, and language modes. The present analysis focused on stops, specifically coronal stops, and recent research proposes that alveolar sounds may enjoy a special status in L2 speech learning due to a universal phonetic bias (Bohn 2020). Thus, future research should focus on other speech segments, particularly those where multiple cues are weighted differently between language pairs, in order to better understand performance mismatches in conjunction with bilingual language modes. The aforementioned avenues of inquiry motivate testable hypotheses that could prove fruitful in the continued development of models of bilingual phonology.

## 5. Conclusions

The present study combined meta-analytic techniques and coronal stop data to assess the extent to which early bilinguals have "compromise" categories for voiceless stops. The results of the analyses (1) suggest that there is little evidence to support this claim in the extant literature, and (2) reinforce the notion that bilingual speech involves dynamic phonetic interactions that can surface as performance mismatches during speech production. The data provide compelling evidence that early bilinguals do not have intermediate, "compromise" phonetic categories, but rather display speech productions that are, by and large, no different from the input of the speech community to which they have been exposed. Taken together, the results support models of bilingual phonology that posit separation between phonological systems and are subject to dynamic phonetic interactions via language activation.

**Supplementary Materials:** The following are available online at https://www.mdpi.com/2226-471 X/6/1/9/s1, All supplementary analyses as well as the data, code, and the experimental materials necessary to reproduce the analyses reported in this article are available at: https://osf.io/un45x/.

**Funding:** This research received no external funding.

**Institutional Review Board Statement:** The study was conducted according to the guidelines of the Declaration of Helsinki, and approved by the Institutional Review Board of the University of Arizona (Approval 12-0250-02).

**Informed Consent Statement:** All participants gave their informed consent for inclusion before they participated in this study.

**Data Availability Statement:** The complete dataset presented in this article is openly available on Open Science Framework and can be accessed at: https://osf.io/un45x/.

**Acknowledgments:** I express my gratitude to Miquel Simonet for sharing the dataset used in the analysis of coronal stops, as well as Kyle Jones for sharing data from his dissertation research. I am grateful for insightful comments from 3 anonymous reviewers that improved the quality of this work. All errors are mine alone.

**Conflicts of Interest:** The author declares no conflict of interest.

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
