# Peer review of "Interlingual Interactions Elicit Performance Mismatches Not “Compromise” Categories in Early Bilinguals: Evidence from Meta-Analysis and Coronal Stops"

_languages, doi:10.3390/languages6010009_

Round 1

Reviewer 1 Report

see attached file.

Reviewer 2 Report

This is an extremely well written paper that will be a great addition to our knowledge of L2 phonological application. I believe that the manuscript needs very minor changes in order to be accepted.

Minor proofreading should be done, specifically:

  • Line 22 > a "compromised" VOT, or, "compromised" VOTs
  • Line 179 > type II errors, or, a type II error
  • Line 193 > based on
  • Line 289 > Figure 2, Amengual, 2011 should be 2012
  • Check most cases of ' "compromise" category' as some may need to be rewritten for grammar

Other aspects

  • Line 70 > Clarify if this quote comes from Williams 1977 or 1980
  • Line 216 > provide citation for this claim
  • Lines 237-8> If I read this correctly, these 20 studies are those listed in Figure 2. If this is the case, state that here.
  • Considering language activation/mode, it should be noted in either the discussion or the conclusion that there are several ways to go about language activation/mode that are not discussed in this paper but should be considered in future research. E.g., cognates (Amengual's 2012 study cited in this study is a great example).

Reviewer 3 Report

Please see attached document with my detailed comments and suggestions for revision

Round 2

Reviewer 3 Report

The author has addressed all comments outlined in the initial review report both comprehensively and effectively. I appreciate the time the author took to carefully respond to and address all comments and concerns raised by reviewers. I have no further comments or requests for edits.

The revised manuscript makes an excellent contribution to the field, and I commend the author for their research and expertise on this topic.